# The Effectiveness of E-Health Interventions Promoting Physical Activity and Reducing Sedentary Behavior in College Students: A Systematic Review and Meta-Analysis of Randomized Controlled Trials

**DOI:** 10.3390/ijerph20010318

**Published:** 2022-12-25

**Authors:** Sanying Peng, Fang Yuan, Ahmad Tajuddin Othman, Xiaogang Zhou, Gang Shen, Jinghong Liang

**Affiliations:** 1Physical Education Department, Hohai University, Nanjing 210024, China; 2School of Educational Studies, University Sains Malaysia, Penang 11800, Malaysia; 3College of International Languages and Cultures, Hohai University, Nanjing 210024, China; 4School of Physical Education, Changzhou University, Changzhou 213164, China; 5Department of Maternal and Child Health, School of Public Health, Sun Yat-Sen University, Guangzhou 510080, China

**Keywords:** e-health, physical activity, sedentary behavior, college student, meta-analysis

## Abstract

Insufficient physical activity (PA) and excessive sedentary behavior (SB) are detrimental to physical and mental health. This systematic review and meta-analysis aimed to identify whether e-health interventions are effective for improving PA and SB in college students. Five electronic databases, including Medline, Web of Science, Embase, Cochrane Library, and ProQuest, were searched to collect relevant randomized controlled trials up to 22 June 2022. In total, 22 trials (including 31 effects) with 8333 samples were included in this meta-analysis. The results showed that e-health interventions significantly improved PA at post-intervention (SMD = 0.32, 95% CI: 0.19, 0.45, *p* < 0.001) compared with the control group, especially for total PA (SMD = 0.34, 95% CI: 0.10, 0.58, *p =* 0.005), moderate to vigorous PA (SMD = 0.17, 95% CI: 0.01, 0.32, *p* = 0.036), and steps (SMD = 0.75, 95% CI: 0.23, 1.28, *p* < 0.001. There were no significant effects for both PA at follow-up (SMD = 0.24, 95% CI: – 0.01, 0.49, *p* = 0.057) and SB (MD = −29.11, 95% CI: −70.55, 12.32, *p* = 0.17). The findings of subgroup analyses indicated that compared to the control group, interventions in the group of general participants (SMD = 0.45, 95% CI: 0.27, 0.63, *p* < 0.001), smartphone apps (SMD = 0.46, 95% CI: 0.19, 0.73, *p* = 0.001), and online (SMD = 0.23, 95% CI: 0.04, 0.43, *p* < 0.001) can significantly improve PA at post-intervention. Moreover, the intervention effects were significant across all groups of theory, region, instrument, duration, and female ratio. At follow-up, interventions in groups of developing region (SMD = 1.17, 95% CI: 0.73, 1.62, *p* < 0.001), objective instrument (SMD = 0.83, 95% CI: 0.23, 1.42, *p* = 0.007), duration ≤ 3-month (SMD = 1.06, 95% CI: 0.72, 1.39, *p* < 0.001), and all female (SMD = 0.79, 95% CI: 0.02, 1.56, *p* = 0.044) can significantly improve PA. The evidence of this meta-analysis shows that e-health interventions can be taken as promising strategies for promoting PA. The maintenance of PA improvement and the effect of interventions in reducing SB remain to be further studied. Educators and health practitioners should focus on creating multiple e-health interventions with individualized components.

## 1. Introduction

Inadequate physical activity (PA) and high levels of sedentary behavior (SB) are crucial risk factors for mortality and non-communicable diseases (NCD), such as cardiovascular disease, cancer, and diabetes around the world [1]. It is well known that regular PA can provide numerous metabolic, cardiovascular, and mental health benefits [2]. Previous studies have also identified that regular PA contributes to prolonged life [3] and promotes students’ academic performance [4]. Despite the variety of health benefits of regular PA, the incidence of SB and physical inactivity (PIA) remains high, particularly during the transition from high school to college, when PA is significantly reduced [5].

Many studies have shown that 40–50% of college students do not participate in enough PA [6,7,8]. China follows the global trend of the prevalence of PIA among college students, which is considered a severe behavioral health risk [9]. According to the survey report by the American College Health Association (ACHA) in 2018 [10], less than one-fourth of students met the PA recommendation (30 min/day for at least five days of the week).

The college stage is critical for forming healthy behaviors [11]. If PIA and SB become habits during this period, it is more likely that this unhealthy lifestyle will be maintained across adulthood, which will bring substantial potential dangers to physical and mental health [11,12]. Therefore, paying attention to the early prevention and effective intervention of college students’ healthy behaviors is necessary.

A recent systematic review identified that psychological factors were among the most critical factors influencing college students’ participation in PA [13]. Effective interventions should focus on vital psychological variables in behavior change. Behavior change techniques (BCTs) such as goal setting, planning, feedback, rewards, and social support have long been essential strategies for improving PA and SB by focusing on interventions that address crucial psychological variables [14,15]. For instance, a recent meta-analysis that included 66 RCTs identified the positive effects of various BCTs in promoting PA in young adults (17–35) [16]. Another review that combined effect sizes measuring SB outcomes also observed a significant reduction in sedentary time in the BCTs intervention group [17]. Although BCTs based on various behavioral science theories have achieved the expected results in improving PA and SB, many face-to-face BCTs interventions (such as group, individual sessions, or counseling) faced implementation dilemmas during the pandemic of COVID-19 [18,19]. At this time, e-health interventions as non-contact or remote techniques are promising strategies that should attract much more attention [20].

E-health (electronic health) refers to information and communication technologies (ICT) related health services, which are delivered or enhanced through electronic devices (e.g., smartphone, computer, pedometer, accelerometer, etc.) and the Internet [21]. Based on this definition of e-health, in this review, e-health interventions were classified into four modalities: smartphone apps, wearable activity trackers (pedometers or accelerometers), social media, and online websites. Previous studies have shown that BCTs based on behavior change theories are the core components of ICT [22,23,24]. E-health is a promising intervention strategy that integrates electronic technology (e.g., various e-health intervention modalities, such as smartphone apps, wearable devices, the Internet, etc.) with psychological factors of behavior change (e.g., goal setting, planning, feedback, rewards, social support, social comparison, etc.) [23]. With the rapid development of electronic technology and the proliferation of smart devices, e-health has become a prominent approach in rehabilitation healthcare and health behavior promotion [23].

E-health interventions have been verified to be effective in a variety of health behavior promotion, including diet [25], weight loss [26], disease management [27], and PA promotion [28,29]. Moreover, the evidence for the effectiveness of e-health interventions for PA and SB is derived mainly from children and adolescents [30,31], patients [32], working women [28], older people [33], and inactive populations [34]. However, there is a shortage of evidence focusing on college students. To our knowledge, only McIntosh et al. have conducted a systematic qualitative review to evaluate the effects of e-health interventions among young people [35]. Although this study found e-health interventions to be effective strategies for increasing PA, the limited number of studies included in the study (*n* = 10) and the lack of some quantitative assessment warranted further exploration of this area. Additionally, there were inconsistent findings of relevant trials conducted among college students. For instance, a study by Al-Nawaiseh et al. found that smartphone app intervention significantly increased college students’ daily steps [36]. In contrast, Epton et al. identified that e-message intervention had no significant improvement in either PA or SB [37]. Based on the research status mentioned above, a quantitative meta-analysis of the existing trials is integral to validating the effects of e-health interventions.

As digital natives, college students have a high penetration of electronic devices and proficient internet skills [38], which can contribute to the widespread application of e-health interventions on campus. Therefore, validating the effectiveness of e-health interventions in increasing PA and reducing SB among college students will provide strong supporting evidence for developing corresponding interventions.

This review is the first meta-analysis to investigate the effects of e-health interventions on promoting PA and reducing SB in college students from a holistic perspective. The purpose of this study was twofold. First, to systematically summarize the effects of e-health interventions for improving PA in terms of total PA (TPA), moderate to vigorous PA (MVPA), light PA (LPA), walking, steps, and SB among college students. Second, to investigate the potential moderators of e-health interventions’ effects through exploratory subgroup analyses of participants’ characteristics and intervention details.

## 2. Materials and Methods

The Preferred Reporting Items for Systematic Reviews and Meta-Analyses (PRISMA) statement [39] and Cochrane Collaboration Handbook recommendations [40] were employed as the rationales and methodological templates of this systematic review. This review has been registered on the PROSPERO platform (CRD42022352623).

### 2.1. Search Strategy

A comprehensive and integrated literature search of randomized controlled trials involving the effects of e-health interventions in improving PA and SB without publication time and language restrictions was conducted for relevant literature published from the following databases: Medline, Web of Science, Embase, Cochrane library, and ProQuest. The search period is from the inception of the databases to 22 June 2022.

Boolean logical operators were used to perform an exhaustive search using the medical topic headings (MeSH) paired with free-text phrases. The leading search terms in the three topics domains are as follows: participants (e.g., college students, university students, tertiary school students); intervention (e.g., e-health, mobile health, smartphone apps, wearable activity trackers, Internet, text messages); outcomes (e.g., physical activity, exercise, sedentary behavior); and study design (e.g., randomized controlled trial, RCT). In addition, as a complementary search, we performed additional screening of top journals (e.g., JMIR Mhealth and Uhealth, Journal of Medical Internet Research, Health Psychology) in the domains of e-health, m-health, and health behaviors to avoid the omission of essential studies due to inclusion criteria. In the Appendix A, specific search information for each database is provided.

All initial search results were imported into Endnote20 software (Thomson ISI Research Soft, Philadelphia, PA, USA). Duplicate studies were removed first. The titles and abstracts of all imported studies were then screened independently by two reviewers to identify the potentially relevant studies that met the inclusion criteria. After the first screening of abstracts and titles, an intuitive and backward snowball retrieval approach was performed to ensure the integrity of the included literature. Then, full-text reviewing was conducted by two reviewers independently to find studies that would be suitable for this review. Regarding disagreement and uncertainty regarding the inclusion of studies, an agreement was reached through consultation with the third reviewer.

### 2.2. Eligibility Criteria

Study eligibility was assessed based on PICOS criteria (participants, interventions, comparators, outcomes, and study design).

#### 2.2.1. Participants

College students who lived alone away from their families were included in this review, which was not limited by gender, age, health status, region, or nationality. Participants were considered eligible if they could participate in the e-health intervention program set up by the researcher. Studies that included university employees among the participants were excluded.

#### 2.2.2. Interventions

Interventions were conducted in college settings. E-health interventions refer to any interventions that include at least one of the following components: smartphone apps; wearable activity tracks; websites; e-messages (i.e., text messages, social media messages, email); telehealth (i.e., remote monitoring, real-time interactive, videotelephony, etc.); or videogame. Studies that comprised multiple group comparisons (i.e., e-health intervention versus multiple interventions) were enrolled, but only the comparisons between the e-health group and the control group were included.

#### 2.2.3. Comparators

Studies were included if neither e-health interventions nor other interventions were imposed in the control groups.

#### 2.2.4. Outcomes

Studies with PA and SB measured with self-report questionaries or objective instruments (pedometers or accelerometers) were included. PA outcomes included TPA, MVPA, LPA, walking, and steps. SB outcomes were the duration of sitting time. PA outcome variables are defined by the individual studies that are included. All the included outcomes should be reported as minutes, hours, or steps per unit. This review also included studies that reported PA in other forms (e.g., energy expenditure, weekly counts, and times per week).

#### 2.2.5. Study Design

Only published RCTs, including pilot RCTs, were considered, while quasi-experiments, cross-sectional surveys, and other qualitative studies were excluded.

### 2.3. Data Extraction

We employed Microsoft Excel to create the data sheets. Two authors (PSY and YF) conducted a separate double-blind investigation to check and extract the crucial information from the included studies. The critical information extracted is as follows: study characteristics (authors, publication year, region); participant characteristics (age, female ratio, health status); intervention details (intervention mode, theory, duration, instrument, outcomes); study design (RCT or not; per-protocol or intention-to-treat; sample size); outcomes of PA and SB. Disagreements in data extraction were resolved through discussion. Missing data were obtained by tracing the literature and emailing the corresponding author of relative studies.

### 2.4. Risk of Bias (ROB) and Quality Assessment

The Cochrane Collaboration risk of bias tool [41] was employed to assess the risk of bias for the included studies using seven domains: (1) random sequence generation, (2) allocation sequence concealment, (3) blinding of participants, (4) blinding of outcome assessment, (5) incomplete outcome data, (6) selective outcome reporting, (7) other undefined biases (such as small sample size and conflict of interest). Each domain was graded as low, unclear, or high risk of bias, respectively, in each study. Each study was classified as low, unclear, or high risk of bias based on a combination of the seven domains. The study was assessed as high risk if more than one item was high risk. If most of the study (over three items) was unclear and there were no high-risk items, the study was assessed as unclear. When there was no high risk or less than three items for unclear, the study was assessed as low risk. Review Manager software (Revman 5.4; The Cochrane Collaboration, The Nordic Cochrane Centre, Copenhagen, Denmark) was used to create the figures of ROB. Two reviewers assessed the ROB of the included studies, and disagreements were resolved by negotiation or consulting the third author.

### 2.5. Statistical Analyses

This review took various outcomes of PA and SB (such as minutes per day or week of TPA, MVPA, and LPA; minutes or times of walking; steps per day; minute per day of sitting time or sedentary time) as the data sources for statistical analyses. The mean (M) and standard deviation (SD) of each outcome at baseline, post-intervention, and follow-up in endpoint were drawn for different calculations of numerical variables to calculate effect sizes based on Cochrane Collaboration Handbook recommendations [40]. First, the effect sizes were pooled using the inverse variance statistical method and random effect models to assess the principal impact of e-health. Standardized mean difference (SMD) representing the pooled effect sizes were supplied, along with 95% confidence intervals (CIs). When pooling the effect sizes of different measurements of PA, each comparison group’s mean and standard deviation were used to determine SMD. If the M and SD were not provided, we performed statistical transformation or requested data from the original authors. Considering the consistency of SB measurements, mean difference (MD) was employed to pool the effect sizes. Second, subgroup analyses of eight moderators conducted in this review are presented as follows: (1) outcome (TPA, MVPA, LPA, walking, steps), (2) participant (inactive vs. general) (inactive refers to participants self-reporting less than 150 min of moderate PA or 75 min of vigorous PA per week; generally refers to participants without any PA level limitation), (3) theory (yes vs. no) (yes: explicitly mentions the adopting of health behavior theories as guidance for the interventions; no: no mention of theories as guidance for interventions), (4) intervention mode (smartphone app, social media, accelerometer or pedometer, online), (5) region (developing vs. developed), (6) instrument (objective vs. subjective), (7) intervention duration or follow-up (>8 weeks vs. ≤8 weeks or >3 months vs. ≤3 months), (8) female ratio (all vs. partial) (all: all participants are females; partial: participants include males). Moreover, *I*^2^ statistics and Cochran Q-test were used to determine the statistical heterogeneity. When *I*^2^ was below 25%, between 25% and 50%, between 50% and 75%, and above 75%, it was classified as very low, moderate, medium, and high heterogeneity, respectively, and *p* < 0.1 for Q test was assessed as statistically significant [42]. To identify publication bias, funnel plots and Egger’s test were adopted [43]. Additionally, a sensitivity analysis was performed to ensure the robustness of the pooled effect size.

All data calculations (such as effect size syntheses, publication bias evaluation, subgroup analysis, heterogeneity tests, and sensitivity analysis) were performed using the statistical software STATA 16.0 (Stata Corp, College Station, TX, USA).

## 3. Results

### 3.1. Literature Search

There were 871 records yielded from the five electronic databases search. Before the screening, 271 duplicates and 126 ineligible records were eliminated. A total of 243 search records reached the next step of screening full texts after carefully reviewing the titles and abstracts of the remaining 474 records. A total of 50 studies were left for full-text review again after 193 records were excluded. At last, this systematic review and meta-analysis included 22 studies [36,37,44,45,46,47,48,49,50,51,52,53,54,55,56,57,58,59,60,61,62,63]. The process of literature selection is shown in Figure 1.

### 3.2. Studies’ Characteristics

All 22 included studies containing 31 effect sizes were conducted on the tertiary education settings in different regions, with 6 studies in developing countries (1 study in China [50], Pakistan [52], Nigeria [53], Sandi Arabia [60,61], Thailand, Chinese Taiwan [56], respectively) and 16 studies (8 studies in the USA [36,44,47,48,49,55,57,59], 3 in Canada [58], 2 in Italy [54,60]; 1 study in South Korea [46], Spain [51], and England [37], respectively) in developed countries. All studies were published in English. The e-health intervention group consisted of 4120 participants, while the control group consisted of 4213 participants, with the mean age ranging between 16 and 27.8. There were 9 studies [44,45,47,52,53,56,59,61,62] exclusively targeting female college students, and 13 studies [36,37,46,48,49,50,51,54,55,57,58,60,63] involved male college students. Detailed information on demographic characteristics is shown in Table A1 (Appendix B).

Regarding the intervention modalities of e-health, we grouped all studies into four categories based on the most dominant interventions in the study. The first is smartphone apps [36,46,48,52,53,58,60,61], which are interventions that mainly set up intervention programs through specific software or deliver intervention content through messages. The second is social media [47,49,50,59], including Facebook, WeChat, etc., where the intervention content was imposed through social media interactions. The third is wearable devices [45,52,54,56,63], such as accelerometers and pedometers, which impose interventions through monitoring and feedback functions on movement. The fourth is online interventions [37,44,55,57,62], which impose interventions through information interactions on specific websites. The duration of the intervention ranged from 1 week to 3 months, and the follow-up period from post-intervention to endpoint ranged from 8 weeks to 15 months. A little over half of the included interventions (12/22, 55%) [37,44,45,48,49,50,51,54,55,56,58,60] were designed based on at least one behavioral theory.

Most of the studies employed a non-intervention control group, and five studies provided their control group with general health information and instructions through sessions [36,48], mental counseling [55], or physical education course [46,56]. The control group of one study [50] needed to report their daily PA duration.

Across all studies, twelve studies [36,45,47,48,49,51,52,53,54,56,61,62] used objective instruments, and ten studies [37,44,46,50,55,57,58,59,60,63] employed self-report questionnaires. When statistical analyses of trials were conducted, the intention-to-treat method was adopted in six studies [48,49,58,59,62,63], and the other studies used values of intervention completers.

### 3.3. Quality of Included Studies

Twenty of the included studies [36,37,44,45,46,47,48,49,51,52,53,55,56,57,58,59,60,61,62,63] had a low or unknown risk of bias, whereas just two studies [50,54] had a high risk of bias. In all 22 studies, sufficient allocation, complete outcome analyses, and reports were observed, whereas only 1 [50] did not conduct random sequence generation. None of the included studies explicitly mentioned blinding of outcome assessment, whereas one study [51] employed blinding of participants. Only one study [54] was identified as having a high-risk bias regarding other biases. Details on both overall and individual quality are shown in Figure 2 and Figure 3.

### 3.4. Primary Outcomes

A meta-analysis of the random effect model including 22 studies (31 effects) yielded a significant improvement in PA in the e-health intervention group at post-intervention compared to the control group (SMD = 0.32, 95% CI: 0.19, 0.45, *p* < 0.001) (see Figure 4), but not a significant improvement at follow-up (SMD = 0.24, 95% CI: −0.01, 0.49, *p =* 0.057) (see Figure 5). At post-intervention, the effect sizes ranged from −0.87 to 1.53; at follow-up, the effect sizes varied between – 0.36 and 1.36. A funnel plot paired with the Egger test (at post-intervention *p* < 0.001; at follow-up *p =* 0.035) indicated that publication bias might be present (see Appendix A, available in the Appendix A).

A meta-analysis of the random effect model for five effect sizes reporting SB-related outcomes found that the e-health intervention did not reduce SB significantly at post-intervention compared to the control group (MD = −29.11, 95% CI: −70.55, 12.32, *p =* 0.17) (see Figure 6). Considering the small number of studies reporting SB, no test for publication bias was performed.

### 3.5. Subgroup Analysis of PA

The subgroup analyses of eight moderator variables at post-intervention and follow-up are shown in Table 1 and Table 2. At post-intervention, the differences in intervention effects were significant in the groups of outcome (*p =* 0.031), region (*p =* 0.046), and instrument (*p =* 0.044). Compared to the control group, interventions from the groups of TPA (SMD = 0.34, 95% CI: 0.10, 0.58, *p =* 0.005), MVPA (SMD = 0.17, 95% CI: 0.01, 0.32, *p =* 0.036), steps (SMD = 0.75, 95% CI: 0.23, 1.28, *p* < 0.001), general participant (SMD = 0.45, 95% CI: 0.27, 0.63, *p* < 0.001), smartphone app (SMD = 0.46, 95% CI: 0.19, 0.73, *p =* 0.001), and online (SMD = 0.23, 95% CI: 0.04, 0.43, *p* < 0.001) can significantly improve PA. Notably, intervention effects were significant across all groups of theory, region, instrument, duration, and female ratio.

At follow-up, the differences in intervention effects were significant in the group of outcome (*p* < 0.001), intervention mode (*p =* 0.031), region (*p* < 0.001), instrument (*p =* 0.005), duration (*p =* 0.043), and female ratio (*p =* 0.049). Compared to the control group, interventions from the groups of steps (SMD = 1.17, 95% CI: 0.87, 1.46, *p* < 0.001), developing region (SMD = 1.17, 95% CI: 0.73, 1.62, *p* < 0.001), objective instrument (SMD = 0.83, 95% CI: 0.23, 1.42, *p =* 0.007), follow-up ≤ 3-months (SMD = 1.06, 95% CI: 0.72, 1.39, *p* < 0.001), and all female (SMD = 0.79, 95% CI: 0.02, 1.56, *p =* 0.044) can significantly improve PA. Additionally, intervention effects were not significant across all groups of participants and theories.

There was no subgroup exploration for the effects of SB due to only five included studies.

### 3.6. Robustness of the Results

Sensitivity analyses were performed to assess the reliability of the results, which were conducted using Stata 16.0. The specific method is to eliminate the literature one by one and then combine the effect sizes to observe whether the results have changed significantly. The sensitivity analysis results showed that the effect sizes did not alter much for PA both at post-intervention and at follow-up, as well as for SB, indicating that the findings of the meta-analysis were robust (see Appendix A, available in the Appendix A).

## 4. Discussion

This systematic review and meta-analysis aim to identify and quantify the valid evidence of the e-health interventions for improving PA and SB among college students. The results indicated that e-health interventions have a significant small-to-moderate effect on PA at post-intervention (SMD = 0.32, 95% CI: 0.19, 0.45, *p* < 0.001) according to Cohen’s criteria [64], whereas the maintenance of PA improvement was not observed because there was no significant effect at follow-up (SMD = 0.24, 95% CI: −0.01, 0.49, *p =* 0.057). Regarding reducing SB, the e-health intervention group contributed to a mean reduction of 29.11 min per day in SB time compared to the control, but the effect was not statistically significant.

The finding of e-health interventions positively affecting increasing PA at post-intervention is powerful support of recently published reviews [28,30,31,32,33,34]. Previous relevant studies mainly focused on populations such as adolescents [30,31], patients [32], women [28], and older people [33], while only this review targeted college students. A meta-analysis by Champion et al. [31] showed that school-based e-health interventions could improve PA in adolescents, and Kwan et al. [33] also found the same effect in older people. Cotie et al. [28] observed large effect sizes in working-age women, and the recent review conducted by Duan et al. [32] has the same findings in NCD patients. Although this review found only small-to-moderate effect sizes of all PA outcomes in college students, the pooled effect size for steps was also close to the large effect size by subgroup analysis of PA outcomes. Based on the extensive validation of e-health interventions’ effects on different populations, such interventions will have a promising prospect of improving PA and SB. Given that different PA outcomes and measures may lead to high heterogeneity, which would impede accurate comparison and interpretation of results [65], the experimental design of future studies should take this into full consideration.

College students have more freedom and independence, often leading to a high risk of developing poor health behaviors due to their lack of self-control and self-efficacy [65,66]. Considering this point, many e-health intervention trials have used self-efficacy as the core theory component [44,48,49,54,55,56,58,60]. Several meta-analyses [32,67,68] also found that theory-based interventions were more effective, which, unfortunately, was not found in this study. Although the college settings and the literacy that college students possess are well-suitable for e-health interventions in improving PA and SB, confusing goal setting and non-targeted intervention content may discourage college students from engaging in some complex PA (e.g., MVPA). This may be one of the reasons for the small effect sizes of MVPA in this review. The design of future e-health interventions should focus on adding specific goals and plans for PA and precisely matching BCTs to goals rather than just general health behavior education or general counseling advice.

High heterogeneities were observed in all three effect estimates. Still, the robustness of the pooled effect size for this review was determined by sensitivity analysis, which also indicates that the results are reliable. Subgroup analysis only found that differences in PA outcomes at follow-up may be a potential source of heterogeneity [69]. Furthermore, in addition to identifying significant intervention effects in many subgroups, we also observed some within-group comparisons of intervention effects where one group had a significantly larger effect size than the other.

From the subgroup analysis of intervention modes, smartphone apps and online interventions had significant intervention effects at post-intervention, with the highest amount of smartphone app intervention effects. Based on the rapid development of technology, smartphones are becoming more and more functional, and the open mobile app development platforms offer many convenient conditions for PA and SB interventions [68,70]. There are more comprehensive information and entire interaction in the interventions through websites, which is perhaps the main reason for their effectiveness. Several studies [34,71,72] confirmed that hybrid intervention modes were better than single intervention modes. However, integrating different intervention modes must be matched with corresponding BCTs so that the key roles’ variables can be easily identified [73,74].

Interestingly, this review found visible differences in the effects of the e-health intervention across different participants. The effect size was larger for the general college students than the inactive college students. The lower level of PA motivation of inactive college students may have impeded the intervention effect [75]. In addition, the intervention effect was better in the group of all female college students, especially at follow-up, where the effect size almost reached the level of a large effect size, which indicated that female college students have better adherence to the e-health interventions. Therefore, the behavioral–psychological characteristics should be fully considered when applying e-health interventions to different participants.

Through subgroup analyses of the two instruments for PA measurement, an important finding of this review was that the objective instrument group had significantly larger intervention effects at both post-intervention and follow-up than the self-report questionnaire group. Self-report is a low-cost, feasible, and convenient method for data collection [76]. Previous studies also verified that the results of self-report questionnaires have high correlations with those of objective instruments [77,78]. However, objective instruments should be promoted to ensure the accuracy and precision of the measurements. Combining accurate algorithms and the portability of measurement tools will facilitate PA-related health behavior studies.

In addition, the effects of two follow-up durations on PA were significantly different, while the difference between the two intervention durations’ effects was marginally significant. The effect size of short-term intervention was larger than that of long-term intervention, which was most pronounced at follow-up. This finding is consistent with Moenninghoff and colleagues’ findings [79]. A possible explanation is that prolonged intervention can lead to losing personal interest and increased objective barriers. Based on this, we suggest that reinforcements should be added to e-health intervention at regular intervals, especially during the follow-up period, which has been verified to be an efficient approach to avoid attenuation [80,81,82].

Regarding SB, this review did not find a significant effect of e-health interventions on SB, which is not sufficient to deny the impact of e-health due to the limited quantity and quality of included studies. This meta-analysis still found a mean reduction of 29.11 min per day in SB after the intervention. A recent meta-analysis by Castro et al. [83] found an increasing trend in sedentary time among college students over the last decade, with an average of 9.82 h per day measured by accelerometers. Therefore, Reducing SB in college students through effective interventions is urgent. To improve the effectiveness of e-health interventions for less SB, providing good monitoring and feedback measures (e.g., setting regular reminders) may be a practical approach [84,85]. Furthermore, future studies should implement trials targeting SB reduction to find key intervention factors that influence SB.

This review is the first meta-analysis to examine the effectiveness of e-health interventions for PA and SB among college students. The included studies are all RCTs that were conducted in over ten countries. The findings of this study can be used as an essential theoretical basis and practical guidance to improve PA and SB among college students. As interventions are highly in tune with intelligent technology, e-health interventions are convenient, efficient, and inexpensive, making them suitable for dissemination and implementation in college settings. Future health promotion projects, especially campus health projects, should employ e-health in their priority list of interventions, which will contribute to the prevention of NCDs and improve the health and well-being of college students.

Despite the innovation and strength of evidence in this study, there are still the following limitations. First, only RCTs were included in this study. Thus, many other relevant trials and investigations have been omitted. Future research should enlarge the search scope to include exhaustive studies for more comprehensive explorations. Second, pooling the effect sizes of different PA outcome variables is a challenging attempt. Although this review has been registered in PROSPERO, the high heterogeneity from numerous potential moderators (i.e., PA outcomes, participant characteristics, intervention modes, durations for intervention implementation and follow-up, and outcome measurements) and not enough included studies contribute to the cautious interpretation of the synthesized results. Third, based on the characteristics of e-health interventions, participant blinding is not possible to perform, which should be the potential reason for downgrading the quality of ROB assessment. Finally, this review has not provided insight into the correlations and mechanisms of action between factors associated with BCTs and the effects of e-health interventions. Further research should focus on these crucial issues.

## 5. Conclusions

The findings of this systematic review and meta-analysis identified that e-health interventions have a significant impact on increasing PA, especially TPA, MVPA, and steps at post-intervention. However, the maintenance of PA improvement at follow-up and the effect of interventions on improving SB remain to be further studied. In addition, the current review provided valuable evidence that the effects of e-health interventions vary in the light of different outcomes and moderators. As promising strategies, e-health interventions have become a new trend in college settings in recent years. Educators and health practitioners should follow this trend and delve into the vital psychological variables of college students’ health behavioral change, integrating smartphone apps, the Internet, monitoring tools, and social media to create multiple e-health interventions with individualized components.

## Figures and Tables

**Figure 1 ijerph-20-00318-f001:**
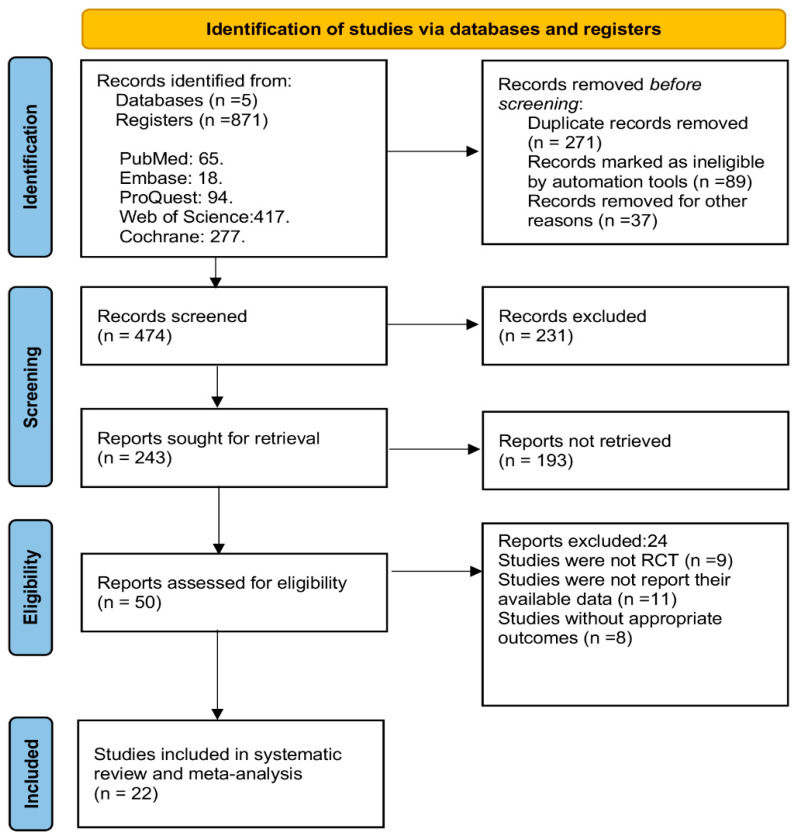
PRISMA flow chart of study selection.

**Figure 2 ijerph-20-00318-f002:**
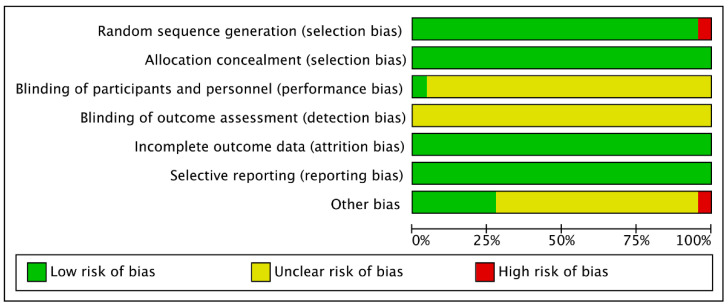
Risk of bias graph (each item presented as percentages).

**Figure 3 ijerph-20-00318-f003:**
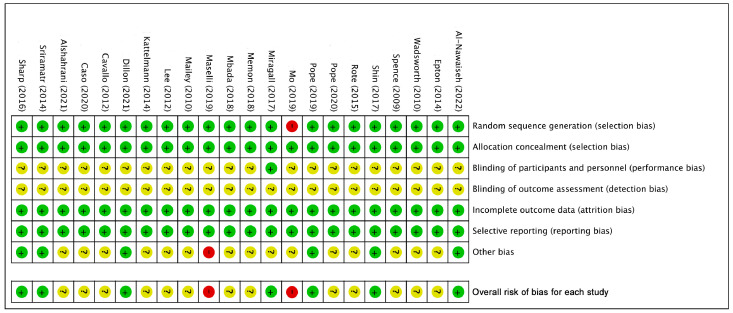
Risk of bias summary for included studies (Green: low risk of bias; yellow: unclear risk of bias; red: high risk of bias) [36,37,44,45,46,47,48,49,50,51,52,53,54,55,56,57,58,59,60,61,62,63].

**Figure 4 ijerph-20-00318-f004:**
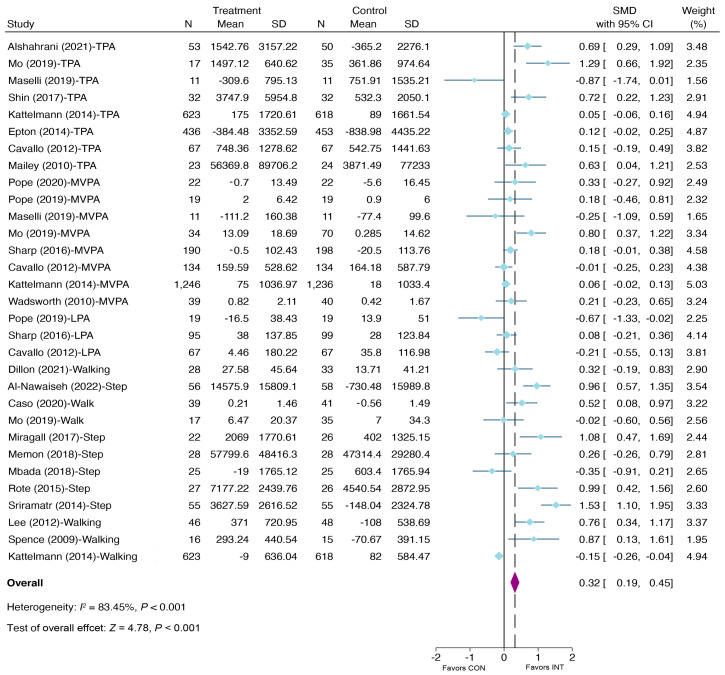
Meta-analysis of effects of e-health interventions on PA at post-intervention versus control [36,37,44,45,46,47,48,49,50,51,52,53,54,55,56,57,58,59,60,61,62,63].

**Figure 5 ijerph-20-00318-f005:**
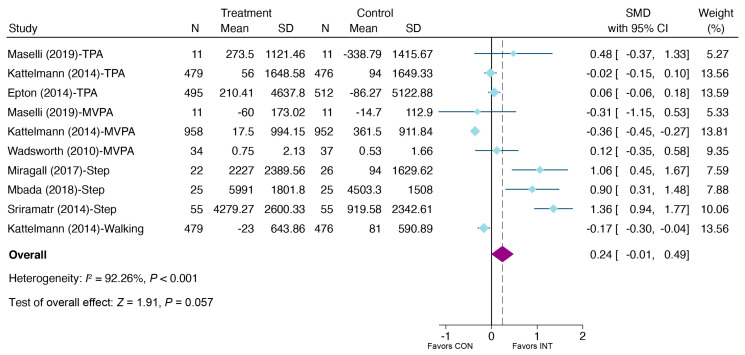
Meta-analysis of effects of e-health interventions on PA at follow-up versus control [37,44,51,53,54,57,62].

**Figure 6 ijerph-20-00318-f006:**
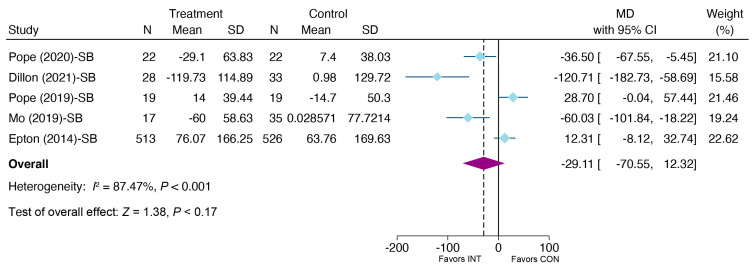
Meta-analysis of effects of e-health interventions on SB versus control [37,48,49,50,58].

**Table 1 ijerph-20-00318-t001:** Subgroup analyses of the effects of e-health interventions on PA at post-intervention.

Moderators	Categories	Studies	Heterogeneity Test		SMD and 95% CI	*p*
			*p*	*I*^2^ (%)		
All studies		31	<0.001	83.4	0.32 (0.19, 0.45)	<0.001
Outcome	TPA	8	<0.001	80.2	0.34 (0.10, 0.58)	0.005
	MVPA	8	0.042	52	0.17 (0.01, 0.32)	0.036
	LPA	3	0.089	58.6	−0.18 (−0.53, 0.18)	0.327
	Walking	6	<0.001	84	0.35 (−0.06, 0.76)	0.092
	Steps	6	<0.001	85	0.75 (0.23, 1.28)	0.005
	Between		0.031			
Participant	Inactive	17	<0.001	77.5	0.19 (−0.03, 0.40)	0.087
	Healthy	14	<0.001	87.1	0.45 (0.27, 0.63)	0.000
	Between		0.066			
Theory	Yes	16	<0.001	72.9	0.36 (0.13, 0.60)	0.002
	No	15	<0.001	87.6	0.28 (0.11, 0.44)	0.001
	Between		0.553			
Intervention mode	Smartphone app	8	0.014	60.2	0.46 (0.19, 0.73)	0.001
	Social media	9	<0.001	80.9	0.26 (−0.07, 0.60)	0.123
	Accelerometer or Pedometer	7	<0.001	75.9	0.31 (−0.04, 0.66)	0.078
	Online	7	<0.001	90.6	0.23 (0.04, 0.43)	0.020
	Between		0.598			
Region	Developing	8	<0.001	82.6	0.63 (0.22, 1.04)	0.002
	Developed	23	<0.001	75.9	0.20 (0.08, 0.32)	0.001
	Between		0.046			
Instrument	Objective	13	<0.001	80.2	0.52 (0.19, 0.85)	0.002
	Subjective	18	<0.001	73.8	0.16 (0.04, 0.27)	0.006
	Between		0.044			
Duration	>8 weeks	18	<0.001	85.7	0.22 (0.06, 0.38)	0.007
	≤8 weeks	13	<0.001	74.4	0.50 (0.25, 0.75)	<0.001
	Between		0.061			
Female ratio	All	11	<0.001	85.5	0.43 (0.10, 0.76)	0.011
	Partial	20	<0.001	80.7	0.26 (0.12, 0.40)	<0.001
	Between		0.349			

**Table 2 ijerph-20-00318-t002:** Subgroup analyses of the effects of e-health interventions on PA at follow-up.

Moderators	Categories	Studies	Heterogeneity Test		SMD and 95% CI	*p*
			*p*	*I*^2^ (%)		
All studies		10	<0.001	92.2	0.24 (−0.01, 0.49)	0.057
Outcome	TPA	3	0.373	0	0.02 (−0.06, 0.11)	0.580
	MVPA	3	0.144	48.4	−0.22 (−0.55, 0.10)	0.171
	Walking	1			−0.17 (−0.30, −0.04)	0.009
	Steps	3	0.414	0	1.17 (0.87, 1.46)	<0.001
	Between		<0.001			
Participant	Inactive	5	0.025	64.2	0.48 (−0.01, 0.96)	0.053
	Healthy	5	<0.001	95.4	0.10 (−0.18, 0.38)	0.500
	Between		0.182			
Theory	Yes	5	0.020	65.7	0.26 (−0.12, 0.64)	0.176
	No	5	<0.001	95.4	0.24 (−0.11, 0.58)	0.178
	Between		0.918			
Intervention mode	Smartphone app	1			0.90 (0.31, 1.48)	0.003
	Social media					
	Accelerometer or pedometer	3	0.034	70.3	0.45 (−0.36, 1.26)	0.275
	Online	6	<0.001	94.3	0.10 (−0.16, 0.36)	0.462
	Between		0.043			
Region	Developing	2	0.206	37.6	1.17 (0.73, 1.62)	<0.001
	Developed	8	<0.001	87.2	−0.00 (−0.20, 0.20)	0.995
	Between		<0.001			
Instrument	objective	4	0.006	75.7	0.83 (0.23, 1.42)	0.007
	Subjective	6	<0.001	87.2	−0.08 (−0.26, 0.11)	0.410
	Between		0.005			
Duration	>8 weeks	6	<0.001	86.5	−0.11 (−0.29, 0.07)	0.232
	≤8 weeks	4	0.259	25.4	1.06 (0.72, 1.39)	<0.001
	Between		<0.001			
Female ratio	All	3	<0.001	86	0.79 (0.02, 1.56)	0.044
	Partial	7	<0.001	88.8	−0.01 (−0.22, 0.20)	0.912
	Between		0.049			

## Data Availability

Data generated or analyzed during this study are included in this published article or in the data repositories listed in the references.

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
