# Peer review of "The Effectiveness of E-Health Interventions Promoting Physical Activity and Reducing Sedentary Behavior in College Students: A Systematic Review and Meta-Analysis of Randomized Controlled Trials"

_ijerph, 2022, doi:10.3390/ijerph20010318_

Round 1

Reviewer 1 Report (Previous Reviewer 1)

Author Response

Dear reviewer,

Thank you for the comments concerning our manuscript entitled “The Effectiveness of E-health Interventions Promoting Physical Activity and Reducing Sedentary Behavior in College Students: A Systematic Review and Meta-analysis of Randomized Controlled Trials” (Manuscript ID: ijerph-1956768). Those comments are all valuable and very helpful for revising and improving our paper, as well as the important guiding significance to our research. We have studied the comments carefully and have made corrections which we hope to meet with approval. Revised portions are marked in highlighted red in the paper. Major corrections in the paper and all point-by-point responses to reviewers' comments are in the attached document.

Reviewer 2 Report (Previous Reviewer 2)

Thanks for giving the chance to review this new version of the manuscript.

Whereas I am satisfied with the authors' revision in terms of changes conducted according to my previous comments, there is a minor point that could be considered for revision. Figure 1. PRISMA flow chart of study selection: Screening section can be divided in two different parts: screening (records screened through title and abstract; and first screening of full text articles) and eligibility (full text article assessed for eligibility). As I assume there is a reason why authors did structure the flow chart in three sections rather than four, I am just suggesting this point so that they can consider the best option fitting the PRISMA guidelines.

Author Response

Dear reviewer,

Thank you for the comments concerning our manuscript entitled “The Effectiveness of E-health Interventions Promoting Physical Activity and Reducing Sedentary Behavior in College Students: A Systematic Review and Meta-analysis of Randomized Controlled Trials” (Manuscript ID: ijerph-1956768). Those comments are all valuable and very helpful for revising and improving our paper, as well as the important guiding significance to our research. We have studied the comments carefully and have made corrections which we hope to meet with approval. Revised portions are marked in highlighted red in the paper. The main corrections in the paper and all point-by-point responses to the reviewer’s comments in the attached document.

Reviewer 3 Report (New Reviewer)

This is a very well done and reported systematic review with metanalysis. It brings relavant contributions for the field.

This is a systematic review with meta-analysis to investigate the effects of e-health interventions on promoting physical activity and reducing sedentary behaviour in college students. The use of health technologies to promote physical activity is a hot topic in the field, evidence synthesis is a desirable topic to be investigated. According to the authors, this is the first meta-analysis that addresses this topic. I confirm that no studies were found in the scientific databases. I suppose that the previous review attended to methodological issues. I have no additional suggestions.

Author Response

Dear reviewer,

Thank you for the comments concerning our manuscript entitled “The Effectiveness of E-health Interventions Promoting Physical Activity and Reducing Sedentary Behavior in College Students: A Systematic Review and Meta-analysis of Randomized Controlled Trials” (Manuscript ID: ijerph-1956768). Those comments are all valuable and very helpful for revising and improving our paper, as well as the important guiding significance to our research. Thank you again for your high approval of our research work, and we will further optimize the manuscript according to other reviewers’ suggestions to meet publication requirements.

Best wishes.

Your sincerely

This manuscript is a resubmission of an earlier submission. The following is a list of the peer review reports and author responses from that submission.

Round 1

Reviewer 1 Report

The article aims to describe the effects of e-health intervention on PA and SB in college students using a systematic review and meta-analysis. While the purpose is novel, the article was difficult to understand due to many grammatical errors in the writing. Another major concern is the selection of subgroups for analysis. For example, the authors chose to perform an subgroup analysis to determine whether there were differences between studies with <100 participants or >100 participants. Why was 100 chosen as the cut point? It seems arbitrary. Additionally, the authors did an analysis between general and inactive student participants. How would you define "general"? The authors also state in their literature search, 193 results were not retrieved. Why were they not retrieve/why were they not accessible? This should also be described as that seems like a very large amount of articles excluded that could possibly fit your inclusion criteria.

This paper does have merit, however, I believe extensive work to improve the grammatical issues and specific definitions of terminology and why decisions were made should be included in the methodology prior to publication of this article. The abstract also needs to include objective data from the results of the meta-analysis.

Reviewer 2 Report

The current study aims to: First, to systematically summarize the effects of e-health interventions for improving PA in terms of total physical activity (TPA), moderate to vigorous physical activity (MVPA), light physical activity (LPA), walking, step, and SB among college students. Second, to investigate the potential moderators of e-health interventions’ effects through subgroup analysis of participants’ characteristics and intervention details. Globally, I commend authors for the rigorous and appropriate systematic review and meta-analyses. Overall, the manuscript is well written, the systematic search seems to be well conducted and the analyses sounds appropriate.

In my humble opinion there are few requirements before the manuscript could be considered for publication.

Abstract

It is not clear what do the authors mean when they refer to "subgroups" throughout the whole abstract.

Lines 23-28. Authors state "There are some important findings... walking and step at follow-up". Please, rewrite the results trying to be accurate enough. It is difficult to understand what the authors mean in terms of specific findings.

Introduction

The structure of the introduction does not help to understand properly the state of the art in terms of the effectiveness of e-health interventions in college students.

For instance, authors mention behavior change techniques but they do not mention how it is related to their study. They showcase interventions in children and adolescents but they do not accurately describe those in college students.

Line 50. Less than one forth?

Lines 58-59. What are the positive effects of the cited BCTs in promoting SB outcomes? Please, provide some examples.

Lines 87-88. Please, rewrite this sentence as the meaning is not clear.

Methods

Line 139. Could be appropriate to change "message" by "e-message"?

Line 201. Please, remove the between before the below.

Results

Appendix A. Table A. What are the specific Apps? Sometimes it is stated the name of the social media used but when referred to smartphone App it is not mentioned the name. In the case it is a customized smartphone App, please, mention it.

Discussion

Lines 355-362. It is not clear enough what is the message/idea of this paragraph. Please, reelaborate the rationale of this paragraph.

Lines 369. It is not clear enough what is the message/idea of this paragraph. Please, re-elaborate the rationale of this paragraph.

Reviewer 3 Report

Dear authors,

the manuscript "The effectiveness of e-health interventions promoting physical activity and reducing sedentary behavior in college students: A systematic review and meta-analysis of randomized controlled trials” included 22 RCTs that examined the effectiveness of e-health interventions for the promotion of physical activity and/or the reduction of a sedentary lifestyle in college students. The manuscript adheres to the PRISMA reporting guidelines, however, due to some methodological concerns (e.g., no protocol registered, study selection process not understandable) my suggestion is to reject the manuscript for publication in IJERPH. In addition, the poor English language makes it very difficult to follow the manuscript – an extensive editing of the manuscript is required.

Abstract: Is not understandable due to poor English language. Avoid abbreviations in the abstract when using them only once.

Introduction: Not clear why the importance of BCTs is described without linking it to e-health interventions (page 2, lines 57-66). Description why e-health interventions are especially for college students suitable is missing (e.g., digital natives). Overview of current literature on e-health PA interventions for college students is insufficient because it focused too much on other populations (e.g., adolescents, patients, elderly) (page 2, lines 70-86). Definition of e-health needs to be stated more clearly (i.e., how are e-health interventions defined within this review) (page 2, lines67-69).

Methods: It is not stated which journals were additional searched for relevant literature (page 3, lines 113-116). The date of the database search and number of hits retrieved is not included in the supplementary material “search strategy”. Regarding the study selection process it is not described whether full-text screening was conducted by two independent researchers (page 3, lines 123-126). An example what is meant by telehealth interventions is missing (page 3, line 140). It is stated that studies are eligible if PA was assessed in TPA, MVPA, LPA, walking and steps. I am wondering about the sentence that PA outcomes in other forms were also included. None of these outcomes are reported in the results section (page 4, lines 150-152). Cut-offs for overall risk of bias assessment seems to be arbitrary and not referring to Cochrane handbook recommendations (page 4, lines 172-176).

Results: The section 3.1 literature search is not understandable without reading the PRISMA flow chart. The PRISMA flow chart is not well structured, e.g., I am wondering why 193 records could not be retrieved for full-text screening. I am not convinced about pooling different PA outcomes that are very heterogeneous in one meta-analysis (figures 4 and 5). For instance, MVPA, LPA and steps differ a lot in the intensity of PA. It would be more reasonable to pool PA outcomes that are similar (see subgroup analysis). Are the total effect sizes of SMD=0.32 (for PA) and SMD=-29.11 (for SB) referring to minutes per day or week? Overall risk of bias for each study should be added to figure 3. Table 1 is not reader friendly.

Discussion: The limitations section (page 13, lines 422-432) is not comprehensive enough. Differences in intensities between PA outcomes should be discussed as well as inclusion of RTCs only, risk of bias assessment (e.g., blinding of participants in e-health PA interventions not possible).